# Collagen Fibril Density Modulates Macrophage Activation and Cellular Functions during Tissue Repair

**DOI:** 10.3390/bioengineering7020033

**Published:** 2020-03-31

**Authors:** Jiranuwat Sapudom, Walaa Kamal E. Mohamed, Anna Garcia-Sabaté, Aseel Alatoom, Shaza Karaman, Nikhil Mahtani, Jeremy C. M. Teo

**Affiliations:** 1Division of Engineering, New York University Abu Dhabi, Abu Dhabi 129188, UAE; jiranuwat.sapudom@nyu.edu (J.S.); wm1081@nyu.edu (W.K.E.M.); anna.sabate@nyu.edu (A.G.-S.); aseel.alatoom@nyu.edu (A.A.); sk6349@nyu.edu (S.K.); nm3489@nyu.edu (N.M.); 2Department of Genetics and Microbiology, Autonomous University of Barcelona, 08193 Barcelona, Spain; 3Department of Mechanical and Biomedical Engineering, New York University, New York, NY 10003, USA

**Keywords:** macrophage, collagen fibril density, immunomechanobiology, monocyte infiltration, fibroblast differentiation

## Abstract

Monocytes circulate in the bloodstream, extravasate into the tissue and differentiate into specific macrophage phenotypes to fulfill the immunological needs of tissues. During the tissue repair process, tissue density transits from loose to dense tissue. However, little is known on how changes in tissue density affects macrophage activation and their cellular functions. In this work, monocytic cell line THP-1 cells were embedded in three-dimensional (3D) collagen matrices with different fibril density and were then differentiated into uncommitted macrophages (M_PMA_) using phorbol-12-myristate-13-acetate (PMA). M_PMA_ macrophages were subsequently activated into pro-inflammatory macrophages (M_LPS/IFNγ_) and anti-inflammatory macrophages (M_IL-4/IL-13_) using lipopolysaccharide and interferon-gamma (IFNγ), and interleukin 4 (IL-4) and IL-13, respectively. Although analysis of cell surface markers, on both gene and protein levels, was inconclusive, cytokine secretion profiles, however, demonstrated differences in macrophage phenotype. In the presence of differentiation activators, M_LPS/IFNγ_ secreted high amounts of IL-1β and tumor necrosis factor alpha (TNFα), while M0_PMA_ secreted similar cytokines to M_IL-4/IL-13_, but low IL-8. After removing the activators and further culture for 3 days in fresh cell culture media, the secretion of IL-6 was found in high concentrations by M_IL-4/IL-13_, followed by M_LPS/IFNγ_ and M_PMA_. Interestingly, the secretion of cytokines is enhanced with an increase of fibril density. Through the investigation of macrophage-associated functions during tissue repair, we demonstrated that M1_LPS/IFNγ_ has the potential to enhance monocyte infiltration into tissue, while M_IL-4/IL-13_ supported fibroblast differentiation into myofibroblasts via transforming growth factor beta 1 (TGF-β1) in dependence of fibril density, suggesting a M2a-like phenotype. Overall, our results suggest that collagen fibril density can modulate macrophage response to favor tissue functions. Understanding of immune response in such complex 3D microenvironments will contribute to the novel therapeutic strategies for improving tissue repair, as well as guidance of the design of immune-modulated materials.

## 1. Introduction

Macrophages are vital innate immune cells with remarkable plasticity that allow them to respond efficiently to pro- and anti-inflammatory signals within their microenvironment [1,2]. Their phenotype and physiology can be altered by the cytokine secretion profile of other cells in the microenvironment in order to modulate the innate and adaptive immune responses [3]. Other than participating in immunological responses, they also play an important role in tissue development, homeostasis, and tissue repair. There is increasing evidence that macrophages in distinct tissues exhibits specific phenotype, activation state and functions [4], therefore it is probable that tissues play an important role in regulating macrophage physiology. Macrophage phenotype has been traditionally divided into 2 groups by simplified classification, namely classically activated macrophages (or pro-inflammatory) and alternatively activated macrophages (or anti-inflammatory). Pro-inflammatory macrophages were described as having inflammatory functions and were crucial for the resistance to pathogens and their elimination, whereas macrophages with anti-inflammatory functions were important for maintaining tissue integrity. However, this simplified nomenclature can create the illusory perception of homogeneity of macrophage subpopulations, which is not shown to exist in vivo, since both cells can phenotypically span the pro-inflammatory–anti-inflammatory spectrum to serve different functions in response to microenvironmental cues [5,6]. While soluble cytokines and cell–cell interactions are well studied, there is still a gap of knowledge on the effects of tissue characteristics as a microenvironmental signal.

Tissue of the cellular microenvironment is dynamic and its density has been shown to be a prominent key player in many physiological and pathological processes, for example wound healing, scar formation and cancer development [7,8,9,10]. Furthermore, it has been demonstrated that tissue repair activities of macrophages are amplified and localized by tissue signals [11,12]. Monocytes are important in these processes; they extravasate into the targeted tissue and differentiate into specific subtype of macrophages to serve the functions that the targeted tissue needs [3]. Many studies, especially in the field of cancer biology, demonstrated that tissue density can modulate cellular invasion and metabolism [7,13,14]. However, less is known on how tissue density modulates monocyte-to-macrophage activation and in turn their phenotypes and functions. Since tissue density is a surrogate for tissue stiffness, a number of studies aimed to study how mechanical properties of materials affect macrophage response [15]. However, these have often been investigated in 2D cell culture models, which may only partially reproduce the complexity of in vivo tissue [16,17]. Although macrophages’ response has been studied in three-dimensional (3D) collagen or fibrin matrices in dependence of ligand presentation [18] and stiffness via crosslinkers [18,19,20], inconsistency in the results has been reported due to the difference of differentiation protocols and cell source used. Additionally, while these studies attempted to demonstrate the important role of tissue mechanical properties on macrophage response, the latter is often investigated based on a number of phenotypic specific markers without taking into account their cellular functions [21]. Despite the growing interest in this field, the direct effect of tissue density on macrophage phenotypes and functions has not been well established. Investigating how tissue density affects macrophage polarization and function allows us to understand how macrophages regulate physiological and non-homeostatic processes, especially in wounds and cancer. Furthermore, it will contribute to the development of novel therapeutic strategies for diseases, as well as guidance of the design of immuno-modulated materials.

This study aimed to investigate the role of tissue/fibril density in the modulation of macrophage phenotypes and functions. Firstly, we embedded monocytic cell line THP-1 into 3D collagen matrices with different fibril density, and differentiated and activated them into specific subtypes of macrophages. The expression of phenotype-specific markers and cytokine secretion profiles were analyzed, characterizing the activation profile of macrophages in dependence of fibril density. To bridge the phenotypical characteristics of macrophages with their cellular functions, we studied their capacity in modulating monocyte infiltration and fibroblast differentiation by co-culturing with THP-1 monocytes and human primary fibroblasts, respectively.

## 2. Materials and Methods 

### 2.1. Preparation and Characterization of 3D Collagen Matrix

Rat Tail type I Collagen (Advanced BioMatrix, Inc. San Diego, CA, USA), 500 mM phosphate buffer (Sigma-Aldrich, St. Louis, MO, USA) and 0.1% acetic acid (Sigma-Aldrich) were used to prepare collagen solution at a concentration of 1 and 3 mg/mL, as previously published [22,23], representing the tissue matrix during the inflammatory and proliferative phases, respectively. THP-1 cells were resuspended in the prepared collagen solution and were transferred onto a glutaraldehyde-coated coverslip, allowing covalent binding of collagen matrix via lysine side chain [24]. The number of THP-1 cells was 1 × 10^5^ per 3D collagen matrix. Collagen fibrillogenesis was initiated at 37 °C, 5% CO_2_ and 95% humidity, embedding cells uniformly in 3D space. Endotoxin levels of collagen was measured to be 0.025 EU/mL and determined that embedded THP-1 were not perturbed by the matrices to invoke an immune response (Appendix A).

Cell-free collagen matrices were analyzed to assess their topological and mechanical properties. Briefly, for topological analysis, the collagen matrices were stained with 50 µM of 5-(and-6)-carboxytetramethylrhodamine succinimidyl ester (TAMRA-SE, Sigma-Aldrich, USA) and visualized using a confocal laser scanning microscope (cLSM) (SP8; Leica, Wetzlar, Germany). The cLSM stacked images were analyzed using a home-built MATLAB script (MATLAB 2019a; MathWorks, USA), as described elsewhere [22,23]. Mechanical properties of cell-free scaffolds were analyzed non-destructively via rheological measurement using ElastoSens Bio 2 (Rheolution, Montreal, QC, Canada). The topological and mechanical characterizations were performed at least in triplicate.

### 2.2. Cell Culture and Macrophages Differentiation

Human monocytic THP-1 cell line (ATCC, Manassas, VA, USA) were cultured in RPMI-1690 media supplemented with 10% fetal bovine serum (FBS), 1% sodium pyruvate, 0.01% of mercaptoethanol and 1% penicillin/streptomycin at 37 °C, 5% CO_2_ and 95% humidity.

Differentiating and activation protocols of THP-1-derived macrophages were adapted and modified from Genin et al. [25]. THP-1 cells embedded in 3D collagen matrices were terminally differentiated into uncommitted macrophages (M_PMA_) with 300 nM phorbol 12-myristate 13-acetate (PMA; Sigma-Aldrich, Germany) in RPMI 1640 media without FBS supplement. After 6 h, differentiating media was removed. Cells were washed with phosphate-buffered saline (PBS) and rested for 24 h in RPMI 1640 without FBS supplement and PMA. Afterwards, cells were activated for 48 h into pro-inflammatory macrophages (M_LPS/IFNγ_) by adding 10 pg/mL lipopolysaccharide (LPS; Sigma, USA) and 20 ng/mL interferon-gamma IFNγ (Biolegend, San Diego, CA, USA), or into anti-inflammatory macrophages (M_IL-4/IL-13_) with 20 ng/mL interleukin 4 (IL-4; Biolegend, USA) and 20 ng/mL interleukin 13 (IL-13; Biolegend, USA).

### 2.3. Quantitative Analysis of Macrophage Cell-Surface Markers

To characterize cells regarding their surface markers, cells were stained with HLA-DRA (subunit of MHC-II) monoclonal antibody (clone: LN3) conjugated with APC-eFluor 780, CD163 monoclonal antibody (clone: eBioGHI/61 (GHI/61)) conjugated with APC, and CD206 (mannose receptor, MMR) monoclonal antibody (clone: 19.2) conjugated with PE for 30 min at standard cell culture conditions. Antibodies were diluted in cell culture medium at a ratio of 1:200. To eliminate the dead cells from the analysis, eBioscience™ Fixable Viability Dye eFluor™ 506 (dilution 1:800) was co-stained with the antibodies. All antibodies and cell viability dye were purchased from eBioscience (eBioscience, San Diego, CA, USA). Stained cells were analyzed using an Attune NxT Flow Cytometer equipped with an autosampler (Thermo Fisher Scientific, Waltham, MA, USA). Compensation settings were applied prior to running the analysis. Cell surface markers are elevated as compared to unstained samples. Unstained macrophages were depicted in Appendix A. Experiments were performed in 4 replicates.

### 2.4. Quantitative Analysis of Pro-Inflammatory Cytokines

To analyze cytokines secreted by macrophages, cell culture supernatants were collected after cell activation. A bead-based multiplex immunoassay for IL-1β, IL-6, IL-8, IL-10, IL-12p70 and TNFα (Biolegend, USA) was utilized to quantify cytokines following instructions by the manufacturer. Samples were analyzed using an Attune NxT Flow Cytometer equipped with autosampler (Thermo Fisher Scientific, USA). Data analysis was undertaken using LEGENDplex™ data analysis software (Biolegend, USA). Experiments were performed in 10 replicates.

### 2.5. Macrophage-Induced Monocytes Infiltration into 3D Collagen Matrices

Prior to performing co-culture experiments, 1 × 10^6^ THP-1 monocytic cells were stained with 1 µM of carboxyfluorescein succinimidyl ester (CFSE; Biolegend, USA) for 15 min at standard cell culture conditions. Afterwards, cells were washed 2 times with PBS and rested in cell culture media. The CFSE staining allowed identification of macrophages from monocytes. 

Macrophages were co-cultured with 1 × 10^5^ CFSE-stained THP-1 monocytic cells for 24 h at standard cell culture conditions. As a control condition, 1 × 10^5^ CFSE stained THP-1 were cultured onto 3D collagen matrices in the absence of macrophages. Afterwards, cells were fixed with 4% paraformaldehyde (Biolegend, USA) for 10 min and permeabilized with 0.1% Triton X100 (Sigma-Aldrich, Germany) for 10 min. Subsequently, cells were washed 2 times with PBS and stained with Hoechst 33342 (dilution 1:10,000 in PBS; Invitrogen, Carlsbad, CA, USA). To quantify monocyte infiltration into 3D collagen matrices, fluorescence stacked images of Hoechst 33342 and CFSE were obtained. Data analysis was performed using a custom-built MATLAB script (MATLAB 2019a; MathWorks, USA), as previously published [22,23]. Percentage of infiltrated monocytes and maximal distance were quantified as a function of macrophage subtypes and tissue density. Experiments were undertaken in 4 replicates at 3 random positions per sample.

### 2.6. Macrophage-Induced Myofibroblast Differentiation in 3D Collagen Matrices

Macrophages were co-cultured with 1 × 10^5^ human primary dermal fibroblasts (ATCC, USA) within the matrix for 3 days in standard cell culture conditions. As negative and positive controls, 10^5^ human primary dermal fibroblasts were cultured onto 3D collagen matrices with and without the presence of 10 ng/mL of transforming growth factor beta 1 (TGF-β1; Biolegend, USA). Afterwards, RNA was extracted and gene expression of myofibroblast markers was quantified using quantitative polymerase chain reaction (qPCR, see gene expression analysis section). Furthermore, as TGF-β1 is involved in the myofibroblast differentiation, we quantified the free active TGF-β1 in the cell culture supernatant using the TGF-β1 ELISA (enzyme-linked immunosorbent assay) kit (Biolegend, USA) following the protocol by the manufacturer. All experiments were undertaken in 4 replicates.

### 2.7. Gene Expression Analysis

Gene expression analysis for myofibroblast differentiation was performed using an established protocol, as published [26]. Briefly, TRIzol (Invitrogen, USA) was used to extract the total RNA. The RNA obtained was converted into complementary DNA (cDNA) using a high-capacity cDNA reverse transcription kit (Applied Biosystems, Foster City, CA, USA). The concentration, and the ratio of absorbance at 260 nm and 280 nm of cDNA were quantified using nanodrops (Thermo Fisher Scientific, USA) prior to performing gene expression analysis. Ribosomal protein S26 (RPS26) and beta-actin were used as a reference gene. The primers were synthesized from Bioneer (Korea). The primer sequences are listed in Appendix A. qPCR was performed using the SYBR Green PCR Master Mix (Applied Biosystems, USA). The qPCR procedure was set as follows: denaturation for 5 min at 95 °C; 45 cycles of denaturation (95 °C, 15 s), annealing under primer-specific conditions (30 s) and target gene-specific extension (30 s at 72 °C). Fluorescence signals were measured for 20 s at 72 °C. To confirm the specificity of the PCR products, melting curve analysis was performed at the end of each run. Experiments were undertaken in 4 replicates.

### 2.8. Statistical Analysis 

Experiments were performed at least in 4 replicates unless otherwise stated. Error bars indicate standard deviation (SD). Levels of statistical significance were determined by a Mann–Whitney test or by one-way or two-way ANOVA followed by Tukey’s using GraphPad Prism 8 (GraphPad Software, San Diego, CA, USA). The significance level was set at *p* < 0.05.

## 3. Results

We postulate that dynamic changes of tissue/collagen fibril density during physiological and pathological processes can modulate macrophage activation and cellular functions significantly. This is especially vital during tissue repair where tissue density changes from loose to dense tissue. To address the underlying questions, we utilized collagen matrices at concentrations of 1 and 3 mg/mL to simulate tissue density differences. Human THP-1-derived macrophages were used for this study since they closely resemble native monocyte-derived macrophages [15,20,25,27]. It is well accepted that monocytes first infiltrate into tissue and then, depending on the microenvironmental cues, they activate into specific subtype of macrophages. To mimic this situation in vitro, THP-1 cells were embedded into 3D collagen matrices, differentiated and activated, as depicted in Figure 1A. 

### 3.1. Characterization of Cell-Free 3D Collagen Matrices

Prior to performing the cell experiment, we characterized our reconstituted 3D collagen matrices in terms of topological and mechanical properties in a cell-free condition. As shown in Figure 1B, it could be observed that fibril density is increased with collagen monomer concentration. The mean pore size was quantified using an image-based toolbox and measured to be 11.07 ± 0.62 µm and 3.74 ± 0.75 µm for 1 mg/mL and 3 mg/mL collagen concentrations, respectively (Figure 1C). The pore size of our 3D collagen matrices are in the range as it is found between 3 to 11 µm in the interstitial tissue [28]. Collagen fibril size was 669 ± 36 nm, similar for both matrices, hence pore size alone is the variable factor in our collagen system here. Elastic modulus of the collagen matrices was inspected using non-destructive rheological measurement, and the mean elastic modulus was measured to be 41.58 ± 9.18 Pa and 211.57 ± 6.79 Pa for 1 mg/mL and 3 mg/mL collagen concentrations, respectively (Figure 1C). The results demonstrated that the elastic modulus increased with higher fibril density, as published elsewhere [22,28]. The thickness of collagen matrices were in the range of 300–400 µm, allowing ample diffusion of nutrients and mediators into the matrices [29].

THP-1 cells were embedded into the well-defined 3D collagen matrices, differentiated and activated into specific subtypes of macrophages, M_PMA_, M_LPS/IFNγ_ and M_IL-4/IL-13_. Cell embedding did not affect the fibrillation and topological properties of the collagen matrix [29,30]. All macrophage subtypes appeared to exhibit a rounded morphology and no matrix-dependent cell morphological differences could be observed (Figure 1D). As is known, pro- (M1) and anti-inflammatory (M2) macrophages could be distinguished by their morphological appearance on 2D cell culture models, but not in 3D fibrillar collagen matrices [18,31]. To demonstrate how specific subtype of macrophages behave in different collagen fibril density, we studied the expression of cell-surface markers and associated genes, cytokines secretion and cell functionality.

### 3.2. Dense Matrix Enhanced Expression of MHC-II, CD163 and CD206 

As shown in Figure 1D and mentioned above, macrophage subtypes could not be distinguished by their morphology when cultured within 3D collagen matrices. To distinguish macrophage subtypes, we quantified our cells with widely accepted macrophage-specific markers, namely MHC-II (through HLA-DR at the protein level, and HLA-DRA at the gene level), CD163 and CD206, using flow cytometry and qPCR after macrophage activation. While upregulation of MHC-II is commonly used as an indicator for increased inflammatory activities via pro-inflammatory antigen-presenting cells, CD163 and CD206 are reserved for the opposite, anti-inflammatory cells [3].

As shown in Figure 2A, M_LPS/IFNγ_ significantly expressed higher MHC-II, a major histocompatibility complex class II, at both protein (Figure 2A(ii)) and gene expression level (Figure 2A(iii)), compared to M_PMA_ and M_IL-4/IL-13_. At the gene level, we found MHC-II tends to be expressed higher in denser tissue, this is significantly pronounced for M_LPS/IFNγ_. This fibril density dependency is not seen at the protein level. A recent report suggested that MHC-II expression is regulated via exogenous mechanical signals by activation of the p38 MAPK pathway [32] which might explain the differential gene expression of MHC-II in our case.

CD163, a scavenger receptor, was uniformly expressed at the cell surface for all our macrophage subtypes (Figure 2B(ii)). However, significant differences in *CD163* gene expression could be observed in M_IL-4/IL-13_ and it also appeared to be tissue density dependent (Figure 2B(iii)), whereby a denser matrix stiffness promotes enhanced expression of CD163 in M_IL-4/IL-13_. Previous studies investigating the expression of CD163 on THP-1-derived macrophages have been conflicting. Tedesco et al. reported that CD163 is undetectable, although Genin et al. revealed no significant difference in CD163 expression between THP-1-derived pro- and anti-inflammatory macrophages [25,33,34], and finally Zhu et al. demonstrated that approximately 20% of THP-1-derived anti-inflammatory macrophages expressed CD163 [27]. All these studies were performed on 2D cell culture plastic. Furthermore, CD163 was found to be hardly presented at the cell surface of peripheral blood mononuclear cell (PBMC)-derived macrophages both in 3D and 2D culture models [30]. It has also been reported that CD163 is expressed in macrophages activated by IL-10 rather than IL-4 in vitro [35].

Next, we analyzed CD206 (Figure 2C), a mannose receptor. No differences in CD206 on the cell surface could be observed in all our macrophage subtypes (Figure 2C(ii)), as previously reported in THP-1-derived pro- and anti-inflammatory macrophages [33]. However, we found upregulation of *CD206* gene expression in dense collagen matrix in all macrophages. The expression level of CD206 was reported to be correlated to collagen morphology, because collagen can exist in the globular or fibrous form at the microscopic level, as reported [36]. This suggests that CD206 expression is sensitive to the biophysical changes of cellular microenvironments. Besides, CD206 plays a role in the cell attachment to collagen via fibronectin type II domain of the CD206 receptor [37] as well as in collagen degradation [38], which might explain the high expression of CD206 in denser collagen matrices. Furthermore, CD206 is reported to be expressed non-specifically in macrophage phenotypes [39].

Overall, we found that MHC-II was expressed higher in M_LPS/IFNγ_ macrophages in dense matrix at the gene level but was similar at the protein level, and higher at both the gene and protein level when compared to other macrophage phenotypes. CD163 was expressed on all macrophages, but it found significantly higher in M_IL-4/IL-13_ at the gene level, and significantly higher in dense matrix. CD206 appeared to be present on all macrophages, but higher in M_LPS/IFNγ_ and M_IL-4/IL-13_ at gene level in dense matrix. Our data suggest the importance of fibril density in regulation of macrophage activation. However, the analysis of surface markers alone is inconclusive for distinguishing between the THP-1-derived macrophage subtypes. It has to be mentioned that in our study, macrophages were differentiated and activated in 3D collagen matrices, thus the expression pattern of surface markers might differ from cells differentiated and activated on 2D surfaces. 

### 3.3. Dense Matrix Enhances Secretion of Cytokines

To further characterize macrophage activation state along with the cell-surface markers, we investigated the secretion of cytokines by our THP-1 derived macrophage subtypes. We measured the secretion level of IL-1β, IL-6, IL-8, IL-10, IL-12p70 and TNFα, during two different biologically relevant scenarios, in the presence of activators and 3 days after the removal of activators in the cell culture media (Figure 3). Removal of activators not only serve to allow us to observe changes in cytokine secretion profiles, moreover, it is important for our co-culture experiments to be void of exogenous activators other than biochemical signals from our macrophage systems. The microenvironment with and void of activators is also physiologically relevant to tissue repair, recapitulating the overlapping inflammatory phase and repair phase, respectively. As shown in Figure 3, M_LPS/IFNγ_ secreted higher levels of IL-1β and TNFα when compared to M_PMA_ and M_IL-4/IL-13_. Except for IL-8, which is undetectable in M_PMA_, the secretion pattern of released cytokines from M_PMA_ was similar to M_IL-4/IL-13_. The secretion levels of IL-10 and IL-12p70 were relatively low in all macrophages with no differences between subtypes, contrary to the secretion levels of the same cytokines from THP-1-derived macrophages on 2D surfaces [15] and 3D freeze-dried and crosslinked collagen scaffolds [20]. We quantified gene expression of IL-10 and found very low expression. The inconsistency of the results might arise from the activation protocol, cell culture dimensionality, scaffold preparation, as well as differences in pore size and elasticity range.

After removal of activators from cells and subsequent culturing for 3 days in fresh cell culture media, IL-10, IL-12 and TNFα became undetectable in the tested media of all macrophages. In addition, IL-1β released is significantly higher in M_-LPS/-IFNγ_ (Figure 3A), while IL-6 is significantly produced in M_-IL-4/-IL-13_ (Figure 3B). IL-6 has both pro-inflammatory and anti-inflammatory effects, pro-angiogenic and regenerative functions [40]. As shown in Figure 3B, the release of IL-6 was very low in the presence of activator in all macrophage subtypes. However, in the absence of activators, enhanced IL-6 production could be observed. The presence of IL-6 after removal of inflammatory cues enhances the further removal of apoptotic cells or debris from the wound site while supporting angiogenetic activities and collagen production required for repair, as published elsewhere [41,42,43,44].

For all cytokines, levels of secretion appear to be fibril density dependent. A dense matrix enhanced most of the cytokine secretion in all macrophage subtypes, both in the presence of activators and after removal of activators. Recent studies demonstrated that THP-1-derived macrophages cultured on soft 2D polyacrylamide hydrogels exhibited reduced production of pro-inflammatory cytokines [15,45,46] via the upregulation of peroxisome proliferator-activated receptor γ (PPARγ) expression [46]. PPARγ has been shown to enhance the differentiation of monocytes into anti-inflammatory macrophages, induced by Th2 cytokines such as IL-4 and IL-13 [47]. However, enhancement of pro-inflammatory cytokines secretion in stiff tissue has been observed in several pathological conditions [48,49,50]. For example, arterial stiffening in cardiovascular diseases is associated with an increase in IL-1β, IL-12, TNFα, and MCP-1 as measured from blood plasma. These clinical findings are in line with our in vitro observations but contradict a report whereby pre-activated (PBMC)-derived macrophages showed lowered pro-inflammatory cytokine secretions in 1-ethyl-3-(3-dimethylaminopropyl)carbodiimide (EDC) crosslinked 3D collagen matrices (stiff substrate) compared to untreated collagen matrices [18]. Enhanced stiffness via increment of fibril density or tuning of collagen fibril bending stiffness using crosslinkers could very well impact different downstream signaling. The exact mechanism of how secretion of pro-inflammatory cytokines is enhanced with an increase of fibril density is still unknown. At this stage we could not provide mechanistic insights into how fibril density regulates cytokine release from macrophages. However, we aim to address this underlying question in future studies.

Overall, we demonstrated that cytokines released from THP-1-derived macrophages in 3D cell culture are fibril density-dependent. The cytokine secretion pattern from macrophage phenotypes is partially similar to cytokines found in 2D cell culture on varying substrate stiffness, as discussed above. Beyond the characterization of macrophages’ phenotypical changes through cell surface markers and induction of cytokines, functional properties of macrophages are attributed to their capacity to produce chemokines, which drive monocytes infiltration, and growth factors that support fibroblast-mediated tissue repair and remodeling.

### 3.4. M_-LPS/-IFNγ_ Enhances Infiltration of Monocytic Cells into 3D Collagen Matrices

Monocyte recruitment from the blood to the injured tissue is essential for the early inflammatory response. We performed a monocyte infiltration assay using CFSE-stained THP-1 cells. Stained cells were co-cultured with different macrophage subtypes for 1 day; this short duration avoids the bias of cell proliferation. Stacked images were gathered using epi-fluorescence microscopy and analyzed using a custom-built image analysis toolbox to reveal cell infiltration into 3D collagen matrices. Representative cross-sectional images of infiltrated CFSE-stained cells in a co-culture condition with macrophage subtypes in different matrix densities are illustrated in Figure 4A. The parameters of cell infiltration, namely the percentage of infiltrated cells and maximum infiltration distance, were analyzed, as previously published [22,23]. Cells found below 20 μm beneath the Coll I matrix surface were considered as infiltrated cells. The maximum infiltration distance was defined as the distance crossed by 10% of all cells. As shown in Figure 4B, in the monoculture or co-culture with M_-PMA_ and M_-IL-4/-IL-13_ approximately 75% of THP-1 cells were infiltrated into 3D collagen matrices, whereby the amount of infiltrated THP-1 cells were increased to 90% when co-cultured with M_-LPS/-IFNγ_. In addition, the presence of M_-LPS/-IFNγ_ enhanced the infiltration distance of infiltrated cells (Figure 4C). However, fibril density did not affect the percentage of infiltrated cells and maximum infiltration distance of THP-1 cells.

Alongside, CX3CL1, MIP1α, and CCL20 [51] monocyte chemoattractant protein-1 (MCP-1, as also known as CCL2) is one of the key chemokines that regulate migration and infiltration of monocytes into tissue [52,53]. Furthermore, MCP-1 has been reported to enhance inflammatory response [54,55] and maintain the pro-inflammatory macrophage phenotype as well [56]. We quantified the secreted amount of MCP-1 in macrophages conservatively in mono-culture conditions in order to validate the chemoattractant potential of monocytes. As shown in Figure 4D, high amount of MCP-1 was significantly secreted by M_-LPS/-IFNγ_, compared to M_-PMA_ and M_-IL-4/-IL-13_. Similar to other cytokines, MCP-1 secretion found to be fibril density dependent and was higher in dense matrix. However, the higher secretion of MCP-1 in a dense matrix did not trigger higher percentage and distance of monocyte infiltration.

Overall, our results demonstrated that M_-LPS/-IFNγ_ are involved in the recruitment and infiltration of monocytes into inflamed tissues. MCP-1 is secreted significantly by M_-LPS/-IFNγ_, rather than M_-PMA_ and M_-IL4/-IL13_, and it can be used for characterization of M_-LPS/-IFNγ_ phenotype. The percentage and distance of infiltrated cells does not correlate with the secretion levels of MCP-1 from M-_LPS-/-IFNγ_.

### 3.5. M_IL-4/IL-13_ Triggers Fibroblast Differentiation into Myofibroblasts

The interplay between fibroblasts and macrophages is the key regulator during the tissue repair phase [57,58,59]. Recent studies addressed this issue through *in vivo* models of tissue injury [60] and mathematical models of cell growth [61]. In this tissue-regeneration stage, fibroblasts differentiate into myofibroblasts in the presence of TGF-β1 [62,63,64,65]. As TGF-β1 is a prerequisite for fibroblast differentiation, we quantified active free TGF-β1 secretion from the supernatant of macrophages in a monoculture using ELISA (Figure 5A). We found a fibril density dependency of TGF-β1 secretion by macrophages. A lower fibril density enhanced the secretion of TGF-β1 in all macrophages, with less significance for M_-IL-4/-IL-13_. However, TGF-β1 secretion was much higher in M_-IL-4/-IL-13_, than M_-PMA_ and M_-LPS/-IFNγ_, suggesting that the M_-IL-4/-IL-13_ might have a potential capability to trigger fibroblast differentiation. The high production of TGF-β1 by M_-IL-4/-IL-13_ is in line with other reports [66,67]. In the co-culture condition with M_-IL-4/-IL-13_, an increased amount of TGF-β1 was detected (Appendix A), indicating that myofibroblasts additively secreted TGF-β1 to maintain fibroblast differentiation in a matrix density dependent manner. A removal of TGF-β1 leads to apoptosis of myofibroblast, as previously reported [68,69]. 

To demonstrate the differentiation of myofibroblastic phenotype, we observed the organization of stress fibers, analyzed myofibroblast marker (alpha smooth muscle actin, aSMA) [70] and the expression of matrix protein, namely alpha 1 chain of type 1 collagen (Coll1a1) and fibronectin containing extra domain A (EDA-FN). As shown in Figure 5B, cells were stained with Phalloidin to visualize stress fibers. In the presence of TGF-β1, pronounced stress fibers could be observed optically, as also reported elsewhere [26,71]. Similar phenotype with pronounced stress fibers could be found in a co-culture condition with M_-IL-4/-IL-13_, but not with M_-PMA_ and M_-LPS/-IFNγ_. Qualitatively, the organization of stress fibers appears to be independent of fibril density. To confirm myofibroblast differentiation in a quantitative manner, we studied the expression of aSMA, Coll1a1 and EDA-FN using qPCR (Figure 5C). The results showed that an increase of aSMA could be found in fibroblasts co-culture condition with M_-IL-4/-IL-13_, but not with M_-PMA_ and M_-LPS/-IFNγ_. The expression of aSMA was solely from myofibroblasts, as aSMA is not expressed in macrophages (Appendix A). Interestingly, aSMA expression decreased with increasing fibril density. In our previous report, we demonstrated that the amount of aSMA-positive cells (investigated by manual counting of aSMA stained cells) are well correlated with the aSMA gene expression [69]. Along with the aSMA expression, we further quantified the expression of matrix proteins, as myofibroblasts are known to highly produce matrix proteins [9,72]. As shown in Figure 5D,E, Coll1a1 and EDA-FN were upregulated together with aSMA upregulation. Again, a slight reduction of both matrix proteins was found with increasing fibril density.

Overall, the results demonstrate that the sensing of fibril density by macrophages modulates TGF-β1 production, which is essential in the tissue repair phase during wound healing. Interestingly, we found that a low concentration of TGF-β1 (300 pg/mL) in the co-culture is needed to trigger myofibroblast differentiation, when compared to the systematic addition of 10 ng/mL TGF-β1. This result is in line with the previous report where slow and sustained TGF-β1 release in a picogram range from polyethylene glycol (PEG) microbeads embedded in 3D collagen matrix could fully differentiate fibroblasts into myofibroblast [73]. We also demonstrated that fibril density could influence myofibroblast differentiation as shown by matrix dependent expression of aSMA and matrix protein secretion (Coll1a1 and EDA-FN) in mono-culture and in co-culture with M_-IL-4/-IL-13_. Dense fibrillar matrix appeared to suppress myofibroblast differentiation. It suggests that a dense fibrillar matrix might act as a negative feedback for myofibroblasts to prevent an excessive production of matrix proteins and remodeling in physiological conditions [74]. Furthermore, the results from the co-culture with fibroblasts supported the observed very low secretion of IL-10 in M_-IL-4/-IL-13_, since the presence of IL-10 in a picogram range could de-differentiate myofibroblast back into fibroblast, as reported in fibroblast mono-culture [69,75,76] and co-culture with macrophages [30].

## 4. General Discussion and Conclusion

The plasticity and cellular functions of macrophages are of interest in many physiologically and pathologically relevant situations. Tissue density plays a key role in modulating phenotypes and functions in many cell types both in vitro and in vivo [13,77,78,79]. However, there exists a lack of knowledge regarding how tissue density affects macrophage activation and functions, especially in the context of normal to cancerous tissue, wound to scar tissue, during tissue repair. Currently, 3D collagen matrices are a versatile platform with adjustable topological and mechanical parameters to replicate interstitial tissue in vitro [7,80,81]. We therefore utilized 3D collagen matrices as a biomimetic tissue model to replicate tissue density.

In this study, human monocytic THP-1 cells were embedded into 3D collagen matrices and activated into distinct macrophage phenotypes in 3D collagen matrices with different fibril density. We demonstrated that M_PMA,_ M_LPS/IFNγ_ and M_IL-4/IL-13_ macrophages expressed phenotype-specific surface markers and have distinct characteristics in terms of cytokine production profiles, triggering monocyte infiltration and differentiation of fibroblasts. M_LPS/IFNγ_ expressed high MHC-II and secreted high amounts of IL-1β, TNFα, MCP-1, whereas M_PMA_ (uncommitted macrophages) expressed no MHC-II and CD163, and secreted similar cytokines to M_IL-4//IL-13_ (high CD163 expression). M_-PMA_, however, demonstrated no capability to differentiate fibroblasts into myofibroblasts when compared to M_-IL-4/-IL-13_. As mentioned in the results, a dense fibrillar matrix enhanced most of the tested cytokine secretion in all macrophage subtypes. At the onset of wound healing, the provisional matrix immediately after wounding consists of a fibrin network, which has been reported to promote a higher pro-inflammatory milieu than collagen [82,83]. Our collagen system mimicked the presence of loose collagen network after the degradation of this fibrin network in the wound bed. The inflammatory response during this transition should be reduced to favor the tissue repair, while an enhancement of inflammatory response in dense tissue can be interpreted in the context of fibrotic tissue and infectious conditions of intact tissues.

Our results essentially confirmed the proposed macrophage phenotypes through functionality assessment in tissue repair context. It has been reported that monocytes infiltrating into tissue could find conditions favorable to becoming an anti-inflammatory phenotype and, thereafter, becoming tissue resident macrophages over time [84], suggesting that M_PMA_ might be conceived as quiescent tissue macrophages [85]. In the absence of activators, cytokine secretion levels were lowered by macrophages, except IL-6, which we measured high concentrations of. We found that denser matrix enhanced cytokine secretion. The underlying molecular mechanism is still unknown and warrants further investigation.

In addition, our results suggest that M_-IL-4/-IL-13_ appears to have a characteristic of the M2a phenotype, similar to other reports that cultured their cells on 2D surfaces [27], since these cells secreted high amounts of TGF-β1 and in turn triggered fibroblast differentiation [67,86]. This hypothesis is also supported by a very low secreted amount of IL-10, which can dedifferentiate myofibroblast back into fibroblasts, as previously reported [69,76]. In addition, a recent report demonstrated that low IL-10 concentration of 120 pg/mL is able to fully de-differentiate myofibroblasts in a co-culture model with macrophages with M2c characteristics [30]. To summarize, the effect of collagen fibril density regulates macrophage-associated functions in a tissue repair context, and a schematic illustration of our findings is depicted in Figure 6. Based on the findings in this study, THP-1-derived macrophages have proven their potential to be implemented as an in vitro tissue repair model, allowing the development of standardized high-throughput platform for anti-fibrosis drug screening and other biomedical studies.

It has to be mentioned that results found in current literature are obtained from different conceptual perspectives and experimental designs. On the one hand, macrophages were pre-differentiated in specific culture conditions, mostly 2D plastic surfaces, and were then harvested and transferred to the testing materials [18]. The conceptual perspective of this experiment, a so-called cell-centric perspective, assumes that a distinct cell phenotype demonstrates a particular response, and it presents the same response even when transferred into the tissue/matrix of different characteristics. On the other hand, monocytes were embedded, differentiated and activated within the testing materials [15,20,46], which is also shown in our study. This conceptual perspective, a so called tissue-centric perspective, assumes that the tissue shapes phenotypes and functions of cells to fulfill the tissue’s need.

To further conclude, there is a growing interest in the use of biomimetic 3D culture systems as tools for understanding tissue-modulated immune functions. The current in vitro cell culture models, although far more simplified than the in vivo models in terms of biochemical and biophysical properties, decouple the necessary parameter of interest, namely tissue/fibril density. As our collagen matrices allow for additional modifications with other ECM components, for example fibronectin [26] and glycosaminoglycans [87,88], as well as a stepwise increase of matrix stiffness via crosslinker [23], we are capable of performing further studies on how matrix characteristics of higher complexities modulate macrophage phenotypes and functions. Additionally, further study will focus on spatio-temporal dynamics of cell–cell interactions during the tissue repair phase at a single cell resolution using label-free cell tracking [89].

## Figures and Tables

**Figure 1 bioengineering-07-00033-f001:**
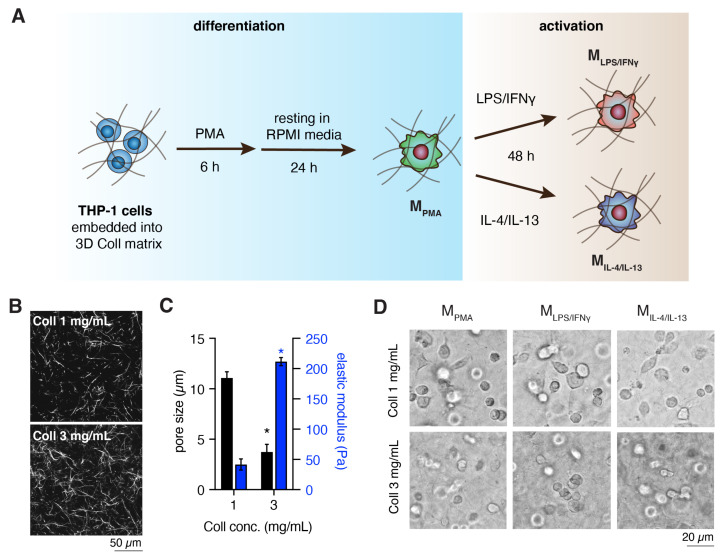
Differentiation and activation of macrophages, matrix characterization and morphology of THP-1 derived macrophages. (**A**) Schematic illustration of macrophage differentiation and activation in 3D collagen matrix. (**B**) Representative images of collagen matrices visualized using confocal laser scanning microscopy (cLSM, scale bar = 50 µm). (**C**) Topological and mechanical characterization of 1 mg/mL and 3 mg/mL collagen matrices. Topological analysis was quantified at 3 random positions of each matrix. All experiments were performed in triplicates. (Data are represented as mean ± standard deviation (SD); * significance level of *p* < 0.05). (**D**) Representative cell morphology of THP-1 derived macrophages in 1 mg/mL and 3 mg/mL of 3D collagen matrices (scale bar = 20 µm).

**Figure 2 bioengineering-07-00033-f002:**
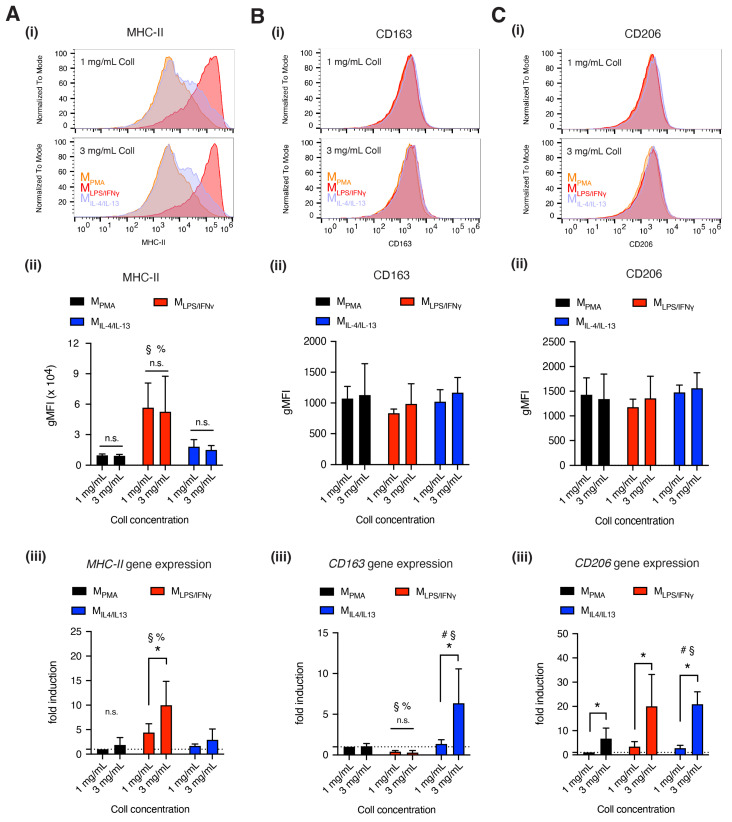
Quantitative analysis of THP-1 derived macrophage surface markers. (**A**) MHC-II, (**B**) CD163 and (**C**) CD206 were analyzed at both the protein level and gene expression. All subfigures, (**i**) depicts the representative histogram plot of stained cells as a function of fluorescence signal intensity. The respective quantitative analysis of geometric mean of the fluorescence intensity is shown in subfigure (**ii**) All experiments were performed at least in triplicate. Gene expression was quantitatively analyzed using quantitative polymerase chain reaction (qPCR) and the results are shown in subfigures (**iii**) Data are represented as mean ± SD; * significance level of *p* < 0.05. The characters #, §, % represent the significance level of *p* < 0.05 when compared to M_PMA_, M_LPS/IFNγ_, and M_IL-4/IL-13_ macrophages, respectively.

**Figure 3 bioengineering-07-00033-f003:**
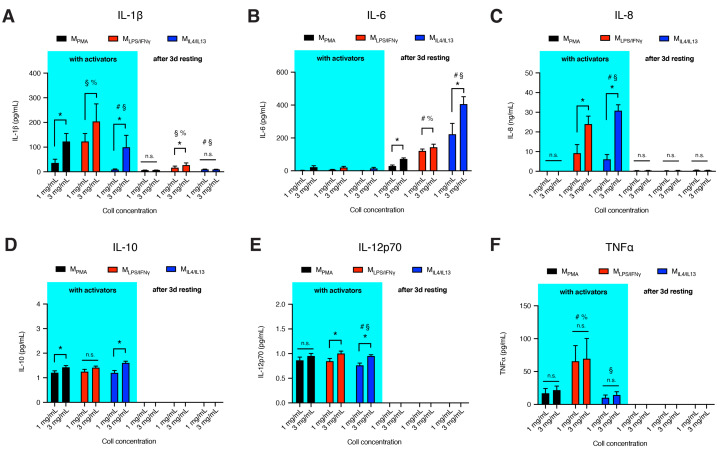
Cytokine secretion by THP-1-derived macrophages in the presence of activators and 3 days after removal of activators. Cytokine secretion (**A**) IL-1β, (**B**) IL-6, (**C**) IL-8, (**D**) IL-10, (**E**) IL-12p70 and (**F**) TNFα. The experiments were performed at least in 4 replicates. Data are represented as mean ± SD; * significance level of *p* < 0.05. The characters #, §, % represent the significance level of *p* < 0.05 when compared to M_PMA_, M_LPS/IFNγ_, and M_IL-4/IL-13_ macrophages, respectively.

**Figure 4 bioengineering-07-00033-f004:**
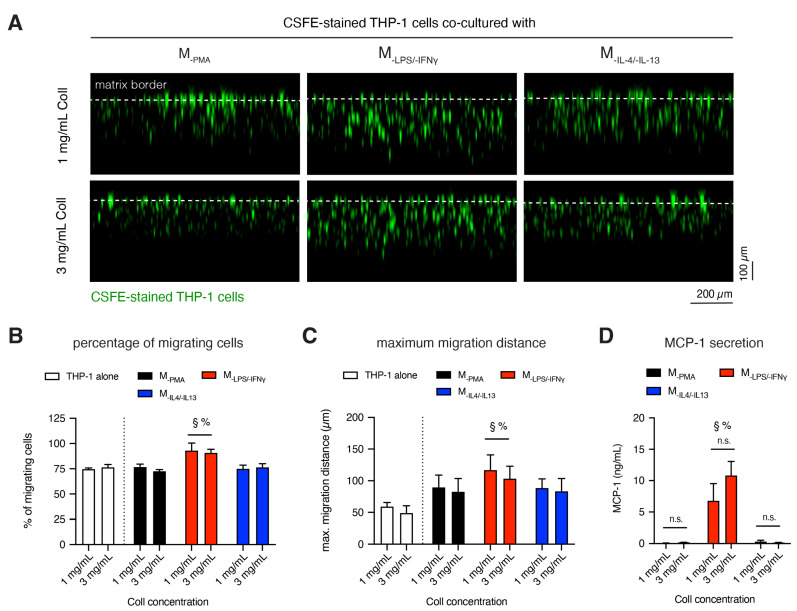
Infiltration of monocytic cells into 3D Coll matrices. (**A**) Representative images of xz-view of carboxyfluorescein succinimidyl ester (CFSE)-stained THP-1 migration into 3D FbColl matrices in the presence of different macrophage subtypes. (**B**) Quantitative analysis of percentage of migrated cells and **(C)** maximum migration distance of CFSE-stained THP-1 cells. Cells found >20 μm beneath the Coll I matrix surface were counted as migrated cells. The maximum migration distance was defined as the distance crossed by 10% of all cells. (**D**) Quantitative analysis of MCP-1 secretion by macrophages in a monoculture condition after 1 day of culture using enzyme-linked immunosorbent assay (ELISA, n = 4). Data are represented as mean ± SD; * significance level of *p* < 0.05. The characters #, §, % represent the significance level of *p* < 0.05 when compared to M_PMA_, M_LPS/IFNγ_, and M_IL-4/IL-13_ macrophages, respectively.

**Figure 5 bioengineering-07-00033-f005:**
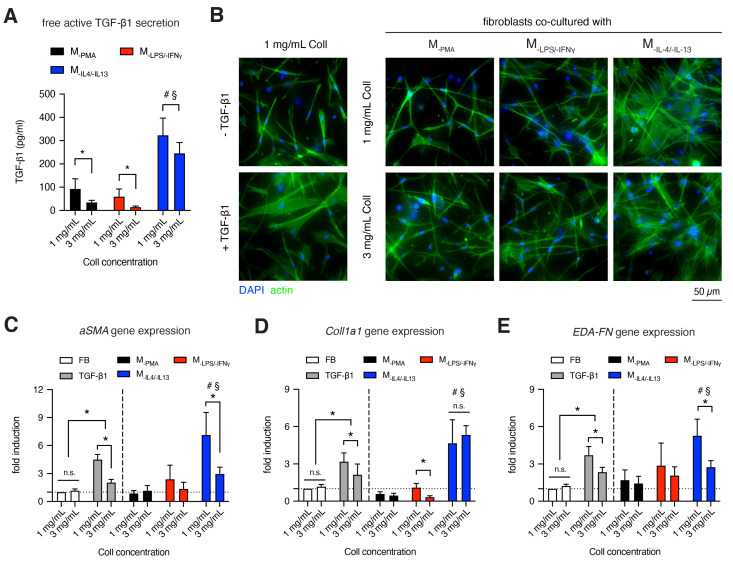
Fibroblast to myofibroblast differentiation in the presence of macrophages. (**A**) Quantitative analysis of active free TGF-β1 secretion by macrophages in a monoculture condition after 3 days of culture using ELISA (n = 6). Human dermal fibroblasts were cultured in a monoculture in the absence/presence of TGF-β1 (white and grey filled bar graphs), or co-culture with macrophages for 3 days (color filled bar graphs). (**B**) Representative images of fibroblasts in monoculture (left column) and co-cultured with macrophages. Cells were stained with Hoechst-33421 and phalloidin conjugated with Alexa Fluor 488 to visualize cell nuclei and f-actin, respectively. Gene expression of (**C**) alpha smooth muscle actin (aSMA), (**D**) Collagen 1 alpha 1 (Coll1a1) and (**E**) fibronectin containing extra domain A (EDA-FN) was analyzed. (n = 4). Fold induction was normalized to fibroblast without TGF-β1 treatment in monoculture. Data are represented as mean ± SD; * significance level of *p* < 0.05. The characters #, §, % represent the significance level of *p* < 0.05 when compared to M_PMA_, M_LPS/IFNγ_, and M_IL-4/IL-13_ macrophages, respectively.

**Figure 6 bioengineering-07-00033-f006:**
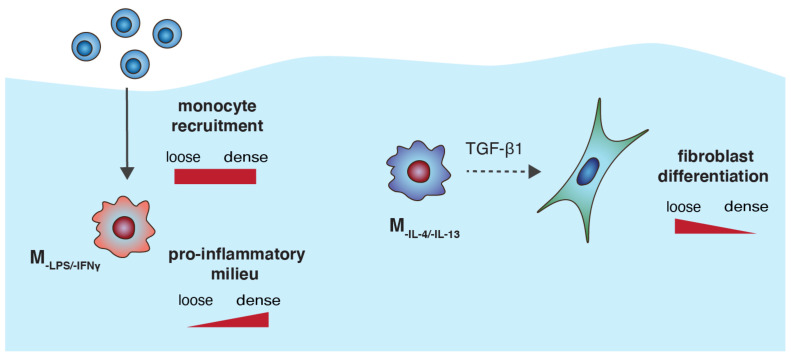
Schematic illustration of the effect of collagen fibril density regulating macrophage-associated functions in a tissue repair context.

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
