# Peer review of "Collagen Fibril Density Modulates Macrophage Activation and Cellular Functions during Tissue Repair"

_bioengineering, 2020, doi:10.3390/bioengineering7020033_

Round 1
Reviewer 1 Report
In the manuscript entitled Collagen fibril density modulates macrophages polarization and cellular functions in wound healing in vitro, Sapudom et al. present an interesting study showing the effects of collagen fibril density on the activation of macrophages driven by the archetypal signals of the macrophage activation paradigm. Furthermore, the study provides insights into the functional consequences of such modulation of macrophage activation by collagen density. There are nevertheless, a few points that require attention to strengthen the study.
Major points
- Figure 2 is hard to interpret because there is a discrepancy between the observations at the mRNA and protein levels. It is my understanding that all the analyses were done 48h post macrophage treatment; could it be that the discrepancy is a result of the dynamic response of the macrophages? It would be worth to do a time course experiment (i.e. 12, 24, 48h) to look at gene and protein induction. Also, how certain are the authors about their cell surface staining –specially CD163 and CD206? It would be important to see a staining control: unstained macs, FMO or isotype control.
- Figure 3 is complicated and it is hard to generalise something from the observations. In my opinion:
- The interpretations regarding IL-6 results are overstated. The pro-inflammatory effects of IL-6 are undermined in the text to fit with the hypothesis.
- The authors assume that IL-6 production can only happen in the absence of activators but again, their observations could be a result of the dynamic response of the macrophages and nothing to do with the presence of the activators. To really conclude differences in the behaviour of macrophages with and without activators, the authors would have to culture their macrophages for three days after removing or not the activators in question. However, this is not the point of the study and I am not sure if the data of rested macrophages is adding anything of value to the story.
- Is it possible that the enhancement of cytokine production by the collagen fibrils is a result of a contamination within the collagen preparations? What happens if the collagen fibrils are completely denatured and added to the macrophages?
- The transition from figure 3 to 4 and 5 is somehow abrupt. Perhaps the authors could explain in the text that beyond phenotypical changes in cell surface molecules and induction of cytokines, functional properties of macrophages are attributed to their capacity to produce chemokines, that drive cellular infiltration, and growth factors and metalloproteases that support tissue repair and remodelling. In accordance with this, they could generate a new figure 4 where they repeat figure 2 but measuring CCL2, TGFb, VEGF, MMP-9 for example? Also, they could validate their observations in PBMC-derived monocytes differentiated to macrophages with M-CSF and subsequently stimulated with the activators used throughout the study.
- To connect their observations regarding CCL2 and the migration assays in figure 4A, the authors could try to block/inhibit CCL2 or its receptor.
- Figure 5 is very interesting. The only caveat is that the induction of TGFb by IL-4/13 treated macrophages, doesn’t match the functional observations regarding the effect of the matrix on the fibroblasts and there are loose ends. Are the fibrils not modulating macrophages but fibroblasts instead? One way to address this is by culturing the fibroblasts with the same concentration of TGFb in matrixes containing collagen at 1 or 3 mg/ml. If the authors do not recapitulate the co-cultures in this system, one way to connect the observations would be to attempt to silence TGFb in their macrophages. In my opinion, the observations in figure 5A could be due to different reasons:
- TGFb has not been measured at the appropriate time? Is the ELISA quantifying activated of total TGFb? Perhaps the difference will become evident if the authors were to measure activated TGFb?
- Alternatively, it could be that the experiments were done with saturating concentrations of IL-4/13? Have these cytokines being titrated?
- The supplemental figure 1 is hard to interpret because TGFb could be expressed by both cell types. Was IL-4/13 added to both cells? If this was the case, is it possible that the difference is now evident in this setting because fibroblasts are competing with macrophages for IL-4/13 availability and basically un-saturating the system?
- It would be important to add a section in the discussion comparing the observations of this study with a physiological scenario. What are the different collagen matrixes resembling? Do they resemble stages of a tissue repair process? If so, would not be expected that macrophages in a loose matrix, resembling a fresh wound, should be more pro-inflammatory? Or is it that the matrixes resemble healthy vs. fibrotic repairing tissue?
Minor points
- The M1/M2(a, b, c) nomenclature should be avoided as proposed in (Murray et al., 2014). It is more appropriate to discuss that PAMPS (such as LPS) and pro-inflammatory factors (typically IFNg) activate macrophages to trigger inflammatory cascades and promote microbial control. On the other hand, anti-inflammatory cytokines like IL-4, IL-10 and TGF-β activate functional properties in macrophages that promote resolution of inflammation and tissue repair. On a similar note, the word polarisation should be changed for activation. The current view is that macrophage activation is not polar but rather modular, therefore the word polarisation is deceiving.
- In the introduction, there are examples of studies addressing how tissue density modulates the biology of other cells however, the authors say that there is still a gap of knowledge on the effects of tissue characteristics as a micro-environmental signal. It would be appropriate to reference here a couple of recent papers showing how the tissue repair activities of macrophages are amplified and localised by tissue signals (Bosurgi et al., 2017; Minutti et al., 2017), as an evidence for what they are proposing to study.
- In line 372, after the phrase: The interplay between fibroblasts and macrophages is the key regulator during the tissue repair, it would be appropriate to cite a couple of recent papers addressing this issue using in vivo models of tissue injury and mathematical models of cell growth (Minutti et al., 2019) and (Zhou et al., 2018).
References
- Bosurgi, L., Cao, Y.G., Cabeza-Cabrerizo, M., Tucci, A., Hughes, L.D., Kong, Y., Weinstein, J.S., Licona-Limon, P., Schmid, E.T., Pelorosso, F., Gagliani, N., Craft, J.E., Flavell, R.A., Ghosh, S., Rothlin, C.V., 2017. Macrophage function in tissue repair and remodeling requires IL-4 or IL-13 with apoptotic cells. Science 356, 1072-1076.
- Minutti, C.M., Jackson-Jones, L.H., Garcia-Fojeda, B., Knipper, J.A., Sutherland, T.E., Logan, N., Ringqvist, E., Guillamat-Prats, R., Ferenbach, D.A., Artigas, A., Stamme, C., Chroneos, Z.C., Zaiss, D.M., Casals, C., Allen, J.E., 2017. Local amplifiers of IL-4Ralpha-mediated macrophage activation promote repair in lung and liver. Science 356, 1076-1080.
- Minutti, C.M., Modak, R.V., Macdonald, F., Li, F., Smyth, D.J., Dorward, D.A., Blair, N., Husovsky, C., Muir, A., Giampazolias, E., Dobie, R., Maizels, R.M., Kendall, T.J., Griggs, D.W., Kopf, M., Henderson, N.C., Zaiss, D.M., 2019. A Macrophage-Pericyte Axis Directs Tissue Restoration via Amphiregulin-Induced Transforming Growth Factor Beta Activation. Immunity 50, 645-654 e646.
- Murray, P.J., Allen, J.E., Biswas, S.K., Fisher, E.A., Gilroy, D.W., Goerdt, S., Gordon, S., Hamilton, J.A., Ivashkiv, L.B., Lawrence, T., Locati, M., Mantovani, A., Martinez, F.O., Mege, J.L., Mosser, D.M., Natoli, G., Saeij, J.P., Schultze, J.L., Shirey, K.A., Sica, A., Suttles, J., Udalova, I., van Ginderachter, J.A., Vogel, S.N., Wynn, T.A., 2014. Macrophage activation and polarization: nomenclature and experimental guidelines. Immunity 41, 14-20.
- Zhou, X., Franklin, R.A., Adler, M., Jacox, J.B., Bailis, W., Shyer, J.A., Flavell, R.A., Mayo, A., Alon, U., Medzhitov, R., 2018. Circuit Design Features of a Stable Two-Cell System. Cell 172, 744-757 e717.
Reviewer 2 Report
This paper details the observations of macrophage differentiation functions using the concentration of collagen as the tested variable. The authors use this 3D model to increase the in vivo-like characteristics of their studies, and provide evidence that the porosity, or collagen concentration, affects the differentiation profile of macrophages. The higher collagen concentration reduces infiltration of monocytes and attenuates the myofibroblast phenotype. The authors use proven methods to differentiate macrophages into pro-fibrotic and pro-inflammatory macrophages, when the cells are embedded within the matrices.
The vibration-based mechanical testing to probe the elasticity showed a low modulus (40 Pa and 200 Pa range). What are the vibration frequencies tested, and what soft materials were used to calibrate the rheometer?
The authors’ use of collagen staining to determine porosity is in line with the literature (reference 25 and (5) below). Have the authors considered measuring the porosity after cell interactions? Are the cells compacting and concentrating the collagen, in other words? If so, this would affect the elastic modulus.
One exciting observation from this study was that the myofibroblast phenotype was attenuated by the increased presence of collagen fibrils. This reviewer was unable to find published literature showing this correlation. The authors used a qualitative stress fiber stain (not described in the methods how this was done) and quantitative qPCR reaction to demonstrate this phenomenon. However, lacking was the alpha-smooth muscle actin (a-sma) staining showing the presence of a-sma localized to stress fibers, which is the definitive evidence for the myofibroblast (1 or 2). Additionally, the phalloidin stain (figure 5B) can be combined with the a-sma stain to provide more data quality (2, 3). The authors explain that they have published a previous paper showing a correlation between qPCR and a-sma stain. Numerous publications have shown this correlation but will also show the immunostain because it can show a single cell’s response to a variable that neither western blot nor qPCR can, and therefore can be quantified (either by confocal or standard fluorescence microscopy). This is especially important because this is a co-culture of cells and is it clear whether the fibroblast or macrophage is expressing a-sma (4)? Please substitute the stress fiber stains with a-sm actin immunostains to provide quantitative and direct myofibroblast presence data.
It is not clear whether the co-culture of macrophages and fibroblasts are within the same collagen gel (perhaps this is the reason for the statements on page 2 line 88 and page 13 line 464); please indicate in the methods (page 4 line 156, “macrophages were co-cultured within the matrix” for example) and page 11, line 401-2 (“in a collagen matrix co-culture condition” for example). If the co-culture allows only media overlap (such as in a Transwell), please indicate in the same places.
Have the authors considered testing whether the cocultures generate different amounts of tissue tension or matrix contraction relative to each other? There should be a logical progression of force generation in these co-cultures that would be interesting and relevant to this field of study. The literature appears to be lacking in these studies. This would probably require a modification of the matrix anchorage used in this study.
References discussed in this review but not in the manuscript:
- Hinz B, McCulloch CA, Coelho NM. Mechanical regulation of myofibroblast phenoconversion and collagen contraction. Exp Cell Res. 2019 Jun 1;379(1):119-128.
- Hinz, B., Celetta, G., Tomasek, J. J., Gabbiani, G. & Chaponnier, C. α-smooth muscle actin expression upregulates fibroblast contractile activity. Mol. Biol. Cell 12, 2730–2741 (2001).
- Vaughan MB, Howard EW, Tomasek JJ. Transforming growth factor-beta1 promotes the morphological and functional differentiation of the myofibroblast. Exp Cell Res. 2000 May 25;257(1):180-9.
- Meng XM, Wang S, Huang XR, Yang C, Xiao J, Zhang Y, To KF, Nikolic-Paterson DJ, Lan HY. Inflammatory macrophages can transdifferentiate into myofibroblasts during renal fibrosis. Cell Death Dis. 2016 Dec 1;7(12):e2495. doi: 10.1038/cddis.2016.402. PMID: 27906172; PMCID: PMC5261004.
- Miron-Mendoza M, Seemann J, Grinnell F. The differential regulation of cell motile activity through matrix stiffness and porosity in three dimensional collagen matrices. Biomaterials. 2010 Sep;31(25):6425-35. doi: 10.1016/j.biomaterials.2010.04.064. PMID: 20537378; PMCID: PMC2900504
Reviewer 3 Report
The manuscript “Collagen fibril density modulates macrophages polarization and cellular functions in wound healing in vitro” by Jiranuwat Sapudom et. al. explores how the density of a collagen lattice affects the macrophages embedded in it. The results are interesting and worthy of being published, but not compelling.
Comments:
The collagen densities chosen have pore sizes which do not restrict cell motion (motion is restricted for sizes lower than 3). Do the results change (particularly the penetration results figure 4) for gels with smaller pore sizes?
Figure 2 why the difference in gene expression but not protein levels?
Figure 2 (and 3, 4, and 5)
Why are you comparing the same cell type with itself A ii) and A iii) SS over cell type M1? It seems like it should be reciprocal. If % is over cell type M1, then SS should be over cell type M2.
Figure 3 has a repeated sentence in the caption and % should be & in the caption?
How do you control for differences in tension in the gels? Do the gels at day 3 have the same tension? Are the results due to fiber density or tension?
Did you check any of the possible differences suggested on page 8 lines 286-291?
Page 9 line 348 “as also known as” get rid of the first as.
Round 2
Reviewer 1 Report
In response to my comments, Sapudom et al. have addressed some issues I raised in my first review. Although, some points were clarified to my satisfaction, there are a few things that still concern me.
- In my original review, I asked if it was possible that the enhancement of cytokine production by the collagen fibrils was a result of LPS contamination within the collagen preparations? I agree with the authors in that denaturating collagen might lead to the change in collagen structure and this could trigger different macrophage responses. However, I strongly disagree with the statement that addressing this issue is beyond the scope of their manuscript. The conclusion from this experiment is central to their story and I am not convinced that their result is not artefactual. What is the equivalent of 0.025EU/ml of LPS in mass/volume? In our lab, we have seen that primary peritoneal macrophages from mouse strongly react to very low concentrations of LPS (as low as 10 fg/ml). What are the levels of IL-1b secreted by THP-1 cells after treatment with 0.025EU/ml of LPS in similar conditions as in figure 3?
- After re-evaluating figure 5 with the clarification of the authors, I feel that the data suggests that collagen fibril mainly modulates fibroblasts and not macrophages. This interpretation is supported by the observation that fibroblast differentiation in response to TGFb in matrixes (without macrophages) containing collagen at 1 or 3 mg/ml, recapitulates their observations using co-cultures. Also, I find surprising that the secretion of active TGFb by M(IL-4/13) in different matrix densities is statistically significant. I think that the comparison should be done using Mann-Whitney test only between them two and not ANOVA. I also think that it would be more honest to show the individual values within the bars to really see the biological difference. In light of this maybe the tone of their interpretations for this section should be more cautious.
- I strongly feel that The M1/M2(a, b, c) nomenclature should be avoided. Please read guidelines from the world experts in macrophage Biology (Murray et al., 2014). They propose that the dogma requires rethinking. In summary, if you call a macrophage M(LPS) and describe its functional characteristics, there is no need to call it M1.
Reviewer 2 Report
All the requested changes have made the manuscript better.
Even though immunocytochemsitry is a better assay to quantify myofibroblasts in a co-culture, the reviewer believes that the authors provided adequate information to conclude that fibroblast activation to myofibroblast phenotype was likely. This will suffice, considering the lag time it would take to perform the best assay during the corona virus pandemic.
There remain a few grammatical errors, including within the new text. Please carefully proofread.
Author Response
All the requested changes have made the manuscript better.
Even though immunocytochemsitry is a better assay to quantify myofibroblasts in a co-culture, the reviewer believes that the authors provided adequate information to conclude that fibroblast activation to myofibroblast phenotype was likely. This will suffice, considering the lag time it would take to perform the best assay during the corona virus pandemic.
We appreciate the reviewer’s understanding with the current situation we are facing.
There remain a few grammatical errors, including within the new text. Please carefully proofread.
We have proofread the manuscript as requested by the reviewer.

Round 3
Reviewer 1 Report
All experimental issues were addressed to my satisfaction. I suggest Sapudom et al. to proof read their manuscript and improve the clarity of the description of their results wherever possible.